# Association between atherogenic index of plasma and depression in premenopausal and postmenopausal women: A cross-sectional study

Siyi Deng[1]☉, Weihong Huang[2]☉, Jinwen Liao[3]☉, Yanling Liu[1], Yang Song[1], Gang Fu[1]*, Mingguo Xu[4]*

1 Department of Cardiology, The Third People's Hospital of longgang, Clinical Institute of Shantou University Medical College (The Third People's Hospital of Longgang District Shenzhen), shenzhen, Guangdong, China, 2 Department of Endocrinology and Metabolism, South China Hospital, Medical School, Shenzhen University, Shenzhen, China, 3 Department of Pediatric, Longgang District Maternity & child Healthcare Hospital of shenzhen city (Longgang Maternity and child institute of Shantou University Medical College), Shenzhen, Guangdong, China, 4 Department of Pediatric, The Third People's Hospital of longgang, Clinical Institute of Shantou University Medical College (The Third People's Hospital of Longgang District Shenzhen), shenzhen, Guangdong, China

☉ These authors contributed to the work equally and should be regarded as co-first authors.
* 18938690175@163.com (MX); fugangxuwei@163.com (GF)

## Abstract

### Aim

This study is the first to investigate the association between the Atherogenic Index of Plasma (AIP) and depression in women, stratified by menopausal status.

### Methods

A total of 9,060 subjects were enrolled from the National Health and Nutrition Examination Survey (NHANES) conducted from 2005 to 2020. AIP was computed by Log10 (triglycerides/high-density lipoprotein cholesterol). Depression was assessed using the Patient Health Questionnaire (PHQ-9), with a score of ≥10 indicating a diagnosis of depression. Multivariate logistic regression, restricted cubic splines (RCS), and subgroup analysis were employed to explore the associations between AIP and depression.

### Results

In comparison to quartile 1, Multivariate logistic regression revealed that AIP in quartiles 2–4 yielded odd ratios (ORs) (95% confidence interval, 95% CI) of premenopausal women of 1.14 (0.87, 1.50), 1.06 (0.79, 1.42), and 1.49 (1.11, 2.00), and postmenopausal women of 0.88 (0.64, 1.22), 1.03 (0.76, 1.41) and 1.40 (1.04, 1.89), respectively. RCS showed a linear correlation between AIP and depression in premenopausal women and a nonlinear correlation between AIP and depression in postmenopausal women. When AIP > 0.60, premenopausal women had an increased risk of depression, while postmenopausal women had a decreased risk of depression.

**Data availability statement:** We uploaded the data to Mendeley Data, and the dataset was assigned a DOI: 10.17632/ykz3983jvz.1

**Funding:** This study was supported by the National Nature Science Foundation of China (81870364 to M.X.), Guangdong Pharmaceutical University Shenzhen Hospital (Longgang) and Guangdong Pharmaceutical University Joint Fund (University joint fund) project (LGSY 202302 to M.X.) and Shenzhen Longgang District science and technology innovation project (LGKCYLWS2022035 to M.X.) and the funders had no role in study design, data collection and analysis, decision to publish, or preparation of the manuscript.

**Competing interests:** The authors have declared that no competing interests exist.

**Abbreviations:** AIP: Atherogenic Index of Plasma; NHANES: National Health and Nutrition Examination Survey; PHQ-9: Patient Health Questionnaire; RCS: restricted cubic splines; ORs: odd ratios; CI: confidence interval; TG: triglycerides; LDL: low-density lipoprotein; HDL: high-density lipoprotein; PIR: poverty-to-income ratio; SD: standard deviations.

## Conclusions

This study demonstrates that elevated AIP levels are significantly associated with an increased risk of depression in both premenopausal and postmenopausal women. However, the strength and direction of these associations varied between the two groups, suggesting that menopausal status may play a critical role in modulating the impact of lipid metabolism on mental health outcomes.

## Introduction

Depression is a prevalent mental health condition characterized by feelings of sadness and uninterested sentiments as well as behavioral, cognitive, or physical symptoms [1,2] and approximately 280 million individuals globally have been diagnosed with depression [3]. It has a substantial negative influence on health, increasing the risk of depression-related suicide and making people more susceptible to illnesses, including cognitive decline [4] and cardiovascular diseases [5,6]. Depression is the leading cause of impairment in the female population [7], with women being nearly twice as vulnerable as men regulated by estrogen [3]. The occurrence of depression in women is associated with changes in the reproductive cycle, including the stages of puberty, postpartum and menopause [8]. Estrogen plays a crucial role in coordinating the metabolism of the female brain and physiological systems. As women approach menopause, decreasing estrogen levels are closely associated with a decline in brain bioenergetics, signaling a shift to a metabolically fragile state [9]. Estrogen deficiency in menopausal women decreases estrogen receptorαexpression in hepatocytes, resulting in altered expression of key enzymes and genes involved in lipid synthesis and catabolism, increases serum levels of total cholesterol, triglycerides (TG), and low-density lipoprotein (LDL), and decreases levels of high-density lipoprotein (HDL) [10]. A growing number of researches are linking disorders of lipid metabolism to depression. A study has shown a significant correlation between elevated levels of HDL-C and TG and depression [11]. Research from experiments on animals suggests that a diet high in cholesterol affects gut microbiota and neuroinflammation, which in turn causes depressive and anxious behaviors in mice [12].

The Atherogenic Index of Plasma (AIP), mathematically expressed as log10 (TG/HDL-C), is closely correlated with lipoprotein particle size [13] and is thought to be associated with the burden of atherosclerosis [14]. Some studies suggest a possible link between high AIP levels and the prevalence of depression [15–17], and the risk of depression is higher in the female population than in the male population [16].

Postmenopausal women are more prone to mood changes due to the altered metabolic state of the body, which may lead to a greater susceptibility to depression. To the best of our knowledge, no prior studies have examined the relationship between AIP levels and depression in women stratified by menopausal status. This study provides a view on how menopausal status may influence the relationship between AIP levels and depression, offering insights into the differing risk profiles for

premenopausal and postmenopausal women and highlighting the importance of considering menopausal status in mental health assessments.

## Methods

### Study population

Data for the study came from the National Health and Nutrition Examination Survey (NHANES) III, a research program conducted by the National Center for Health Statistics(NCHS), Centers for Disease Control and Prevention designed to assess the health and nutritional status of adults and children in the U.S. NHANES utilizes a complex, multistage, probability sampling approach to obtain data through interviews, questionnaires, mobile medical examinations, and laboratory tests [18].

The study protocol conforms to the ethical guidelines of the 1975 Declaration of Helsinki as reflected in a priori approval by the NCHS Ethics Review Board, and all study subjects provided written informed consent. Moreover, the data are available on NHANE's official website https://wwwn.cdc.gov/nchs/nhanes/analyticguidelines.aspx.

This study analyzed data from 2005–2020 and included a total of 38,623 individuals. Individuals aged < 18 years, pregnant or breastfeeding women, missing the data on depression symptoms, menopausal or pregnancy, and missing data on HDL or TG were excluded, and 9,060 subjects were finally included and analyzed. The flow chart of the study is illustrated in Fig 1.

### Menopause state and AIP

Based on self-reported questionnaire answers about reproductive health, women were classified as menopausal or postmenopausal based on whether they reported having had a hysterectomy or a menopause/change of life within the last 12 months.

The Atherogenic Index of Plasma (AIP) is calculated using indicators derived from blood samples, specifically by applying a base-10 logarithmic transformation to the ratio of triglycerides (TG) to high-density lipoprotein cholesterol (HDL-C), both expressed in molar concentrations (mmol/L) $[\log 10(TG/HDL)]$ [19].

In the NHANES program, blood samples are handled according to standardized procedures that include placing the sample on ice immediately after collection, then separating the serum/plasma and freezing it at −20°C before transporting

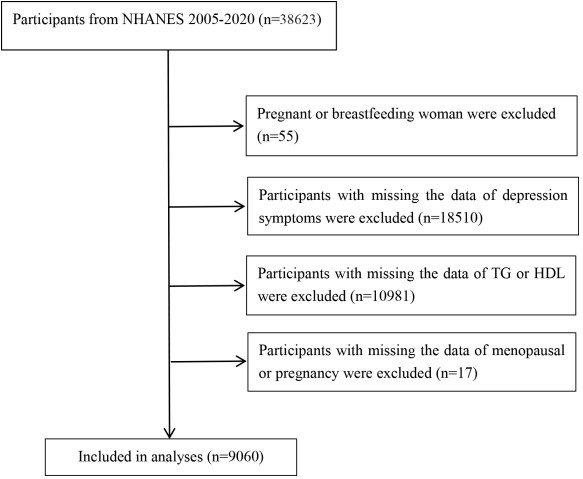

**Fig 1. Flow chart of study participants.** Abbreviations: TG, triglycerides; HDL, high-density lipoprotein.

it to the National Center for Environmental Health (NCEH) laboratory. All laboratory measurements were performed using the Roche Cobas 6000 Chemiluminescent Immunoassay platform and the manufacturer's quality control system and test kits and following U.S. laboratory accreditation standards.

Serum TG is measured using an enzymatic assay based on lipase hydrolysis and subsequent oxidation of phospho-glycerol. HDL-c is measured using a direct immunosuppression method in which an antibody directed against β-lipoproteins selectively permits HDL-c measurements while blocking other lipoproteins. Detailed laboratory procedures and quality control protocols can be found on the NHANES website.

## Depression

The Patient Health Questionnaire-9 (PHQ-9) was employed by the researchers to evaluate depression. The PHQ-9, which contains nine items, is a self-reporting instrument for screening depression, which conforms to the criteria outlined in the *Diagnostic and Statistical Manual of Mental Disorders (Fifth Edition)*. This technique evaluates how frequently depression has occurred throughout the last two weeks. When there are no depression symptoms, a score of 0 is given, and when symptoms happen daily, a score of 3 is given. The ultimate score on the questionnaire ranges from 0 to 27. A cutoff score of 10 or greater indicates the presence of depression, with 88% specificity and sensitivity [20].

## Covariates

Data were collected from the NHANES household interviews, including age, height, weight, race/ethnicity, alcohol consumption, smoking status, marital status, education level, household poverty-to-income ratio (PIR), diabetes, and hypertension. Mexican American, Other Hispanic, Non-Hispanic White, Non-Hispanic Black, and Other Race-Including Multi-Racial were the five categories used to classify race and ethnicity. Alcohol consumption was considered drinking≥12 drinks per year, with no alcohol consumption considered drinking<12 drinks per year [21]. Non-smokers were defined as those who smoked fewer than 100 cigarettes in their lifetime. Those who had smoked at least 100 cigarettes were categorized as current smokers if they continued to smoke, and as former smokers if they had quit smoking [22]. Marital status was categorized as married, separated, including widowed and divorced groups, or never married [23], and educational level was categorized as less than high school, high school or equivalent, college or above. Household PIR were categorized into 0–1.0, 1.1–3.0, and >3.0 [24]. The criteria for diabetes were glycosylated hemoglobin (HbA1c)≥6.5%, fasting plasma glucose≥7.0 mmol/L, using diabetes medication, or a medical professional's previous diagnosis [24]. Hypertension was based on blood pressure of at least 140/90 mmHg, receiving medications for hypertension, or a self-reported history of diagnosed hypertension [18]. Clinical indicators like HDL and TG were measured in the NHANES laboratory.

## Statistical analysis

By the NHANES analysis guidelines, all analyses considered a complex survey design, including sample weights, clustering, and stratification. Missing values for covariates were supplemented using multiple interpolations. Study subjects were assigned quartiles (Q1-Q4) based on the AIP level, with continuous variables being presented as weighted means±standard deviations (SD) for weighted characterization, while categorical variables were expressed as frequencies with weighted percentages. Differences in covariates across the groups were performed using one-way ANOVA for continuous variables and the Rao-Scott chi-square test for categorical variables.

In investigating the relationship between AIP and depression, the population is divided based on premenopausal and postmenopausal status for subgroup analysis. Logistic regression models were used to calculate odds ratios (ORs) and 95% confidence intervals (CIs) for depression. In order to make the statistical results more robust, we constructed multivariate models by adjusting for potential confounding variables as much as possible based on relevant literature evidence. Model 1 was unadjusted, Model 2 was adjusted for age, race/ethnicity, smoking status, and alcohol consumption, and

Model 3 was adjusted for age, race/ethnicity, smoking status, alcohol consumption, education level, marital status, household PIR, hypertension, and diabetes.

Restricted cubic splines (RCS) were used to model the non-linear association between AIP and depression. The analyses were stratified by alcohol consumption, smoking status, marital status, educational level, diabetes, and hypertension history.

All statistical analyses were carried out using R software (version 4.2.3) to account for the NHANES complex sample design using a two-sided test, and $P < 0.05$ was regarded as statistically significant.

## Results

### Baseline characteristics of the study subjects

The baseline characteristics of the study subjects stratified by the AIP quartiles are shown in Table 1. A total of 9060 women were enrolled in this study, with 4715 identified as premenopausal women and 4345 as postmenopausal women. The PHQ-9 scores were used to divide the participants, and 1003 women (9.2%) showed symptoms of depression. Subjects were classified into four groups by the AIP quartiles as follows: Q1 ($-1.25 \leq AIP < -0.34$), Q2 ($-0.34 \leq AIP < -0.14$), Q3 ($-0.14 \leq AIP < 0.08$), Q4 ($0.08 \leq AIP < 1.79$). Subjects with higher AIP were more inclined to be older, non-Hispanic White, drinkers, smokers, married, and with higher levels of education.

### Association of depression with AIP

Table 2 shows the occurrence of depression. Multivariate logistic regression models were built to examine the association between depression and AIP levels stratified by menopausal status. From lowest to highest AIP quartile, the multivariate-adjusted ORs and 95% CIs were 1.00 (reference), 1.14 (0.87, 1.50), 1.06 (0.79, 1.42), and 1.49 (1.11, 2.00), respectively, for premenopausal women; 1.00 (reference), 0.88 (0.64, 1.22), 1.03 (0.76, 1.41) and 1.40 (1.04, 1.89), respectively, for postmenopausal women, in Model 3 after adjusting for age, race/ethnicity, smoking status, alcohol consumption, education level, marital status, household PIR, hypertension and diabetes. As the AIP quartiles increased in all subjects, the risk of depression in premenopausal and postmenopausal women increased (both $P$ trend $< 0.001$).

We conducted multivariable-adjusted RCS analyses to evaluate the non-linear relationship between the AIP levels and depression (Fig 2). RCS exhibited a linear connection between AIP and depression in premenopausal women ($P$ for non-linearity $> 0.05$) and a non-linear connection between AIP and depression in postmenopausal women ($P$ for non-linearity $< 0.05$). The AIP level associated with the lowest risk of depression was $-0.47$ in the overall population. The value of OR had an initial steep increase when AIP ranged from $-0.47 \sim 0.26$, then plateaued. The risk of depression increased as AIP levels increased in premenopausal women. The risk of depression increased when AIP ranged from $-0.45 \sim 0.18$, and there was a trend for decreasing the risk of depression when AIP $> 0.60$ for women who were in a postmenopausal state.

### Subgroup analysis

Subgroup analyses and interaction tests were based on variables including alcohol consumption, smoking status, married status, education level, diabetes, and hypertension (Table 3). We did not observe interactions between AIP and these variables ($P > 0.05$ for interaction).

## Discussion

Using a simple and easily available indicator, AIP, to represent the degree of atherosclerosis, we conducted a cross-sectional study to reveal, for the first time, the relationship between AIP levels and depression in women in different menopausal states, and our results suggest that high AIP levels are associated with a high risk of depression in premenopausal and postmenopausal women. No significant difference was seen in the risk of depression between premenopausal and

**Table 1. Characteristics of premenopausal and postmenopausal women.**

| Variable | Quartiles of AIP | | | | | P-value |
|---|---|---|---|---|---|---|
| | Total | Q1 | Q2 | Q3 | Q4 | |
| **Age (years)** | 47.94 ± 17.58 | 44.64 ± 17.64 | 46.90 ± 17.88 | 48.89 ± 17.48 | 51.48 ± 16.54 | **<0.001** |
| **Race and ethnicity (%)** | | | | | | **<0.001** |
| Mexican American | 1426(7.8) | 223(5.6) | 312(6.8) | 422(9.2) | 469(9.7) | |
| Other Hispanic | 946(5.6) | 176(4,8) | 238(5.7) | 250(5.8) | 282(6.0) | |
| Non-Hispanic White | 3728(68.0) | 836(64.8) | 908(67.7) | 929(67.9) | 1055(71.6) | |
| Non-Hispanic Black | 2002(11.5) | 748(16.8) | 572(13.1) | 440(10.3) | 242(5.6) | |
| Other/Multiracial | 958(7.2) | 285(8.1) | 232(6.8) | 222(6.8) | 219(7.2) | |
| **Alcohol consumption (%)** | | | | | | **<0.001** |
| Yes | 5287(66.5) | 1433(71.9) | 1339(66.6) | 1297(66.2) | 1218(61.2) | |
| No | 3773(33.5) | 835(28.1) | 923(33.4) | 966(33.8) | 1049(38.8) | |
| **Smoking status (%)** | | | | | | **<0.001** |
| Never smoker | 5109(52.5) | 1355(55.7) | 1324(54.2) | 1295(54.4) | 1135(45.6) | |
| Ever smoker | 2968(36.9) | 613(30.8) | 696(35.3) | 721(35.3) | 938(46.5) | |
| Current smoker | 983(10.6) | 300(13.5) | 242(10.4) | 247(10.3) | 194(7.9) | |
| **Married status (%)** | | | | | | **<0.001** |
| married | 4312(54.0) | 1019(54.3) | 1022(51.2) | 1138(55.5) | 1133(55.0) | |
| separated | 2755(25.4) | 651(22.2.) | 683(25.9) | 682(24.9) | 739(28.6) | |
| never married | 1993(20.6) | 598(23.5) | 557(22.8) | 443(19.6) | 395(16.4) | |
| **Educational levels (%)** | | | | | | **<0.001** |
| less than high school | 2024(14.8) | 314(8.7) | 437(12.9) | 562(16.4) | 711(21.4) | |
| high school or equivalent | 1977(22.4) | 401(15.8) | 499(23.7) | 530(23.7) | 547(26.7) | |
| college or above | 5059(63.8) | 1553(75.5) | 1326(63.5) | 1171(59.9) | 1009(51.9) | |
| **Household PIR (%)** | | | | | | **0.022** |
| 0-1.0 | 2028(15.1) | 429(12.3) | 507(15.7) | 525(15.3) | 567(17.1) | |
| 1.1-3.0 | 6955(84.0) | 1816(86.5) | 1740(83.5) | 1721(83.8) | 1678(82.1) | |
| >3.0 | 77(1.0) | 23(1.2) | 15(0.8) | 17(0.9) | 22(0.9) | |
| **BMI (kg/m²)** | 29.25 ± 7.60 | 25.72 ± 6.27 | 28.15 ± 7.13 | 30.62 ± 7.60 | 32.67 ± 7.53 | **<0.001** |
| **TG (mmol/l)** | 1.30 ± 1.03 | 0.61 ± 0.17 | 0.94 ± 0.20 | 1.33 ± 0.29 | 2.36 ± 1.56 | **<0.001** |
| **HDL (mmol/l)** | 1.54 ± 0.44 | 1.94 ± 0.45 | 1.62 ± 0.32 | 1.43 ± 0.28 | 1.17 ± 0.25 | **<0.001** |
| **Menopausal (%)** | | | | | | **<0.001** |
| Pre | 4715(52.8) | 1435(63.0) | 1231(55.4) | 1098(50.6) | 951(41.9) | |
| Post | 4345(47.2) | 833(37.0) | 1031(44.6) | 1165(49.4) | 1316(58.1) | |
| **Depressive symptoms (%)** | | | | | | **<0.001** |
| Yes | 1003(9.2) | 194(6.7) | 214(7.5) | 238(9.6) | 357(13.4) | |
| No | 8057(90.8) | 2071(93.3) | 2048(92.5) | 2025(90.4) | 1910(86.6) | |
| **Diabetes (%)** | | | | | | **<0.001** |
| Yes | 1581(13.7) | 178(5.1) | 281(9.0) | 443(14.9) | 679(26.2) | |
| No | 7479(86.3) | 2090(94.9) | 1981(91.0) | 1820(85.1) | 1588(73.8) | |
| **Hypertension (%)** | | | | | | **<0.001** |
| Yes | 3707(37.7) | 656(24.5) | 827(33.4) | 1037(42.2) | 1187(51.0) | |
| No | 5353(62.3) | 1612(75.5) | 1435(66.6) | 1226(57.1) | 1080(49.0) | |

Data are shown as the mean ± SD or frequency (percentage). Abbreviations: AIP, Atherogenic index of plasma; SD, standard deviation; PIR, poverty-to-income ratio; BMI, body mass index; TG, triglycerides; HDL, high-density lipoprotein.

Q1:1st quartile; Q2: 2nd quartile; Q3: 3rd quartile; Q4: 4th quartile.

Statistical significance (P < 0.05) is indicated by boldface.

**Table 2. Multivariate regression analysis of AIP with depression stratified by menopausal status.**

| Variable | Model 1 | | Model 2 | | Model 3 | |
|---|---|---|---|---|---|---|
| | OR(95%CI) | *P*-value | OR(95%CI) | *P*-value | OR(95%CI) | *P*-value |
| Total | | | | | | |
| AIP | 2.43(1.98,2.99) | **<0.001** | 2.39(1.93,2.97) | **<0.001** | 1.76(1.40,2.21) | **<0.001** |
| Q1 | ref | | ref | | ref | |
| Q2 | 1.12(0.91,1.37) | 0.287 | 1.11(0.90,1.36) | 0.322 | 1.02(0.83,1.26) | 0.849 |
| Q3 | 1.26(1.03,1.53) | **0.025** | 1.26(1.03,1.54) | **0.028** | 1.09(0.88,1.34) | 0.434 |
| Q4 | 2.00(1.66,2.41) | **<0.001** | 1.95(1.61,2.38) | **<0.001** | 1.53(1.25,1.88) | **<0.001** |
| *P* for trend | | **<0.001** | | **<0.001** | | **<0.001** |
| Premenopausal | | | | | | |
| AIP | 2.27(1.70,3.03) | **<0.001** | 2.24(1.65,3.05) | **<0.001** | 1.65(1.18,2.29) | **0.003** |
| Q1 | ref | | ref | | ref | |
| Q2 | 1.28(0.96,1.70) | 0.095 | 1.29(0.96,1.72) | 0.090 | 1.14(0.87,1.50) | 0.352 |
| Q3 | 1.26(0.95,1.69) | 0.109 | 1.27(0.95,1.71) | 0.105 | 1.06(0.79,1.42) | 0.689 |
| Q4 | 1.95(1.49,2.55) | **<0.001** | 1.93(1.45,2.56) | **<0.001** | 1.49(1.11,2.00) | **0.007** |
| *P* for trend | | **<0.001** | | **0.003** | | **0.035** |
| Postmenopausal | | | | | | |
| AIP | 2.54(1.88,3.43) | **<0.001** | 2.29(1.68,3.12) | **<0.001** | 1.62(1.16,2.24) | **0.004** |
| Q1 | ref | | ref | | ref | |
| Q2 | 1.08(0.81,1.45) | 0.604 | 1.04(0.77,1.40) | 0.813 | 0.88(0.64,1.22) | 0.452 |
| Q3 | 1.51(1.15,2.00) | **0.004** | 1.45(1.09,1.93) | **0.010** | 1.03(0.76,1.41) | 0.834 |
| Q4 | 2.15(1.65,2.80) | **<0.001** | 1.98(1.50,2.60) | **<0.001** | 1.40(1.04,1.89) | **0.028** |
| *P* for trend | | **<0.001** | | **<0.001** | | **0.014** |

Model 1: unadjusted

Model 2: adjusted for age, race/ethnicity, smoking status, and alcohol consumption.

Model 3: adjusted for age, race/ethnicity, smoking status, alcohol consumption, education level, marital status, household PIR, hypertension, and diabetes. Abbreviations: ref, reference; AIP, Atherogenic index of plasma;

Q1:1st quartile; Q2: 2nd quartile; Q3: 3rd quartile; Q4: 4th quartile.

Statistical significance (*P*<0.05) is indicated by boldface.

postmenopausal women, but there was a linear correlation between AIP and depression in premenopausal women and a nonlinear correlation between AIP and depression in postmenopausal women. When AIP>0.60, premenopausal women had an increased risk of depression, while postmenopausal women had a reduced risk of depression.

Depression is a complex neurological disorder, and there is growing evidence that fluctuations in estrogen levels in women significantly affect the risk of developing depression [25]. Perimenopausal women have a high incidence of depression in women due to the significant changes in estrogen levels in their bodies as ovarian function begins to diminish [26,27]. As an important point in a woman's life, menopause needs more research to focus on this stage, and finding factors associated with depression in women and intervening in advance is crucial to protecting women's mental health. AIP is closely related to the size of lipoprotein particles [13], representing the atherosclerotic burden, with higher AIP levels indicating greater atherosclerosis [14]. A previous study showed that high AIP levels were significantly associated with a high risk of depression in the general population [16]. To the best of our knowledge, no studies have been conducted to elucidate the relationship between AIP and depression by dividing the female population by menopausal status. This study is the first to investigate the relationship between AIP and depression in a large sample of women with different menopausal status.

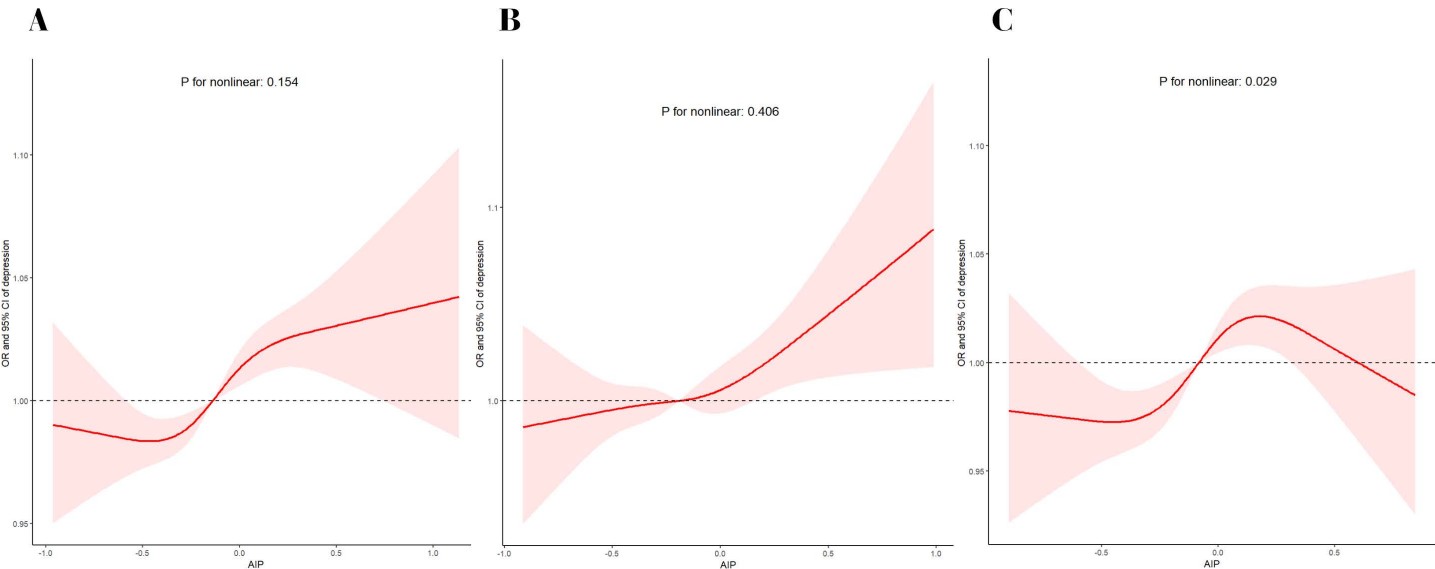

**Fig 2. Association between AIP and depression in total people (A), premenopausal women (B) and postmenopausal women (C).**

Our study showed that high AIP levels were associated with a high risk of developing depression in premenopausal and postmenopausal women. The vasopressor hypothesis suggested that atherosclerosis predisposes subjects to depression [28], and it had been suggested that frontostriatal and frontolimbic dysfunction was the underlying cause [29]. White matter lesions associated with atherosclerosis may lead to ischemia in brain tissue associated with emotion regulation [30,31]. Interaction between depression and atherosclerosis: atherosclerosis can lead to depression through a variety of mechanisms, including dysfunction of the autonomic nervous system, hyperactivity of the hypothalamic-pituitary-adrenal axis [32], increased platelet activation and hypercoagulability [33], altered inflammatory responses, and depression contributes to the development of atherosclerosis through mechanisms such as elevated cortisol levels and impaired endothelial function [33].

However, no significant differences in AIP levels and risk of depression were seen between premenopausal and postmenopausal women [1.49 (1.11,2.00) vs. 1.40 (1.04,1.89)]. The results of an Australian study showed no statistically significant difference in the occurrence of clinical depression between premenopausal and postmenopausal women [23,25,26]. According to estrogen withdrawal theory, it is hypothesized that dramatic fluctuations in hormones can lead to the onset or worsening of depression. The dramatic decrease in estrogen occurs primarily during the perimenopausal period, while hormone levels level off in premenopausal and postmenopausal women, which may partially explain the lack of a significant difference in the risk of depression between premenopausal and postmenopausal women. However, more research is still needed to explore the mechanisms involved.

Our study showed a linear correlation between AIP and depression in premenopausal women, with the risk of depression increasing with increasing levels of AIP. This is similar to the study by Tiemeier et al., who observed a dose-response relationship between atherosclerosis and depression, with more severe atherosclerosis associated with more severe depressive symptoms [34]. Of note, the present study found a nonlinear correlation between AIP and depression in postmenopausal women. Postmenopausal women have a reduced risk of depression when AIP > 0.60. This is contrary to previous findings on atherosclerosis and depression, and according to the estrogen withdrawal theory, the likely reason for this is that estrogen levels in postmenopausal women decline significantly and fluctuate almost imperceptibly, and their effect on depression is stronger than the effect of atherosclerosis on depression. Another possible reason lies in the

**Table 3. Subgroup analysis of AIP and depression.**

| Variable | Premenopausal women | | | Postmenopausal women | | |
|---|---|---|---|---|---|---|
| | OR (95%CI) | *P*-value | *P* for interaction | OR(95%CI) | *P*-value | *P* for interaction |
| **Alcohol consumption** | | | 0.378 | | | 0.750 |
| Yes | 1.67(1.11,2.52) | 0.014 | | 1.27(0.81,1.97) | 0.299 | |
| No | 1.47(0.84,2.58) | 0.176 | | 2.02(1.24,3.27) | 0.004 | |
| **Smoking status** | | | 0.890 | | | 0.963 |
| Never smoker | 2.08(1.28,3.39) | 0.003 | | 1.36(0.83,2.25) | 0.233 | |
| Ever smoker | 1.26(0.76,2.09) | 0.374 | | 1.90(1.12,2.91) | 0.016 | |
| Current smoker | 1.46(0.52,4.12) | 0.471 | | 1.61(0.55,4.76) | 0.389 | |
| **Married status** | | | 0.515 | | | 0.720 |
| married | 1.72(0.97,3.06) | 0.054 | | 1.78(1.03,3.09) | 0.040 | |
| separated | 2.30(1.27,4.15) | 0.006 | | 1.38(0.88,2.17) | 0.158 | |
| never married | 1.08(0.60,1.94) | 0.790 | | 2.04(0.68,6.11) | 0.201 | |
| **Educational levels** | | | 0.160 | | | 0.532 |
| less than high school | 1.41(0.73,2.72) | 0.306 | | 1.85(1.07,3.20) | 0.028 | |
| high school or equivalent | 1.23(0.59,2.54) | 0.585 | | 0.94(0.47,1.90) | 0.866 | |
| college or above | 1.95(1.23,3.11) | 0.005 | | 1.78(1.05,3.02) | 0.032 | |
| **BMI** | | | 0.908 | | | 0.725 |
| <24 | 1.25(0.55,2.82) | 0.594 | | 1.88(0.81,4.34) | 0.139 | |
| >24 | 1.82(1.25,2.66) | 0.002 | | 1.57(1.10,2.26) | 0.014 | |
| **Diabetes** | | | 0.725 | | | 0602 |
| Yes | 1.95(0.78,4.82) | 0.152 | | 1.52(0.88,2.63) | 0.131 | |
| No | 1.59(1.11,2.26) | 0.011 | | 1.68(1,12,2.54) | 0.013 | |
| **Hypertension** | | | 0.096 | | | 0.453 |
| Yes | 2.50(1.32,4.75) | 0.005 | | 1.88(1.27,2.78) | 0.002 | |
| No | 1.40(0.95,2.07) | 0.086 | | 1.00(0.55,1.82) | 0.993 | |

limitations of the study sample. In the population of postmenopausal women, the sample size of AIP > 0.6 was only 67, and the large difference in sample size has a non-negligible impact on the results. Due to the specificity of the postmenopausal period, the relationship between AIP and depression in the postmenopausal female population seems to be more complex compared to premenopausal women. In future studies, we should incorporate both estrogens as well as indicators of atherosclerosis to further explore the mechanisms of depression in postmenopausal women.

Inevitably, there are some limitations to this study. Firstly, this was a cross-sectional study and could not verify a causal relationship between AIP and depression. Secondly, although we attempted to adjust for a variety of potential confounders, limited by the database, this study was unable to adjust for all factors that may affect depression and lipid levels, for example, whether the subject has been treated with antidepressants and is taking medications that affect blood lipid levels, and there was residual bias. Thirdly, because complete years of estrogen data are not available in the NHANES database, we were unable to verify the relationship between estrogen levels and AIP at different menopausal states and whether estrogen levels affect the relationship between AIP and depression. Finally, the results of this study are based on data from the US population, and the conclusions drawn may not apply to populations in other regions.

## Conclusion

In the female population, the degree of atherosclerosis was significantly associated with depression. AIP was linearly correlated with depression in premenopausal women and nonlinearly correlated with depression in postmenopausal women.

AIP can be used as an index of depression in women and provide a reference for healthcare professionals to pay attention to the mental health of women in different menopausal states.

## Acknowledgments

The authors express their gratitude to the participants and staff of the NHANES for their invaluable contributions to this study.

## Author contributions

**Conceptualization:** Siyi Deng, Weihong Huang, Jinwen Liao, Yanling Liu, Yang Song.

**Data curation:** Siyi Deng, Weihong Huang, Yanling Liu, Yang Song.

**Funding acquisition:** Mingguo Xu.

**Investigation:** Siyi Deng, Weihong Huang, Jinwen Liao, Yanling Liu, Yang Song.

**Methodology:** Siyi Deng, Weihong Huang.

**Project administration:** Weihong Huang, Gang Fu, Mingguo Xu.

**Resources:** Siyi Deng, Gang Fu, Mingguo Xu.

**Software:** Siyi Deng, Weihong Huang, Yang Song, Mingguo Xu.

**Supervision:** Jinwen Liao, Gang Fu, Mingguo Xu.

**Validation:** Jinwen Liao.

**Writing – original draft:** Siyi Deng, Weihong Huang, Jinwen Liao, Yanling Liu, Yang Song.

**Writing – review & editing:** Siyi Deng, Weihong Huang, Gang Fu, Mingguo Xu.

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
