## [Decision Letter · Decision Letter 0]

22 May 2025

Dear Dr. Ming-Guo Xu,

Thank you for submitting your manuscript to PLOS ONE. After careful consideration, we feel that it has merit but does not fully meet PLOS ONE’s publication criteria as it currently stands. Therefore, we invite you to submit a revised version of the manuscript that addresses the points raised during the review process.

We look forward to receiving your revised manuscript.

Kind regards,

Emmanuel Kwaku Ofori, PhD

Academic Editor

PLOS ONE

Journal Requirements:

“This study was supported by the Guangdong Basic and Applied Basic Research Foundation (2022A1515012468) and Shenzhen Longgang District science and technology innovation project (LGKCYLWS2022035).”] 

“This study was supported by the Guangdong Basic and Applied Basic Research Foundation (2022A1515012468) and Shenzhen Longgang District science and technology innovation project (LGKCYLWS2022035).”

“This study was supported by the Guangdong Basic and Applied Basic Research Foundation (2022A1515012468) and Shenzhen Longgang District science and technology innovation project (LGKCYLWS2022035).”

Additional Editor Comments (if provided):

Thank you for choosing PLOS ONE. The manuscript has been reviewed and a decision of revision is required. Please address all editorial and reviewer comments in your revision.

General comments

An easy to follow paper worth publishing. Authors should however improve on the discussion part finding plausible explanations to support the results obtained or otherwise. There are few situations in text that need to be supported by citation/references.

Abstract

First sentence is quite misleading and repetitive. There are some few studies that has explored AIP and depression in postmenopause (see PMID: 39648880 ....... there are few others). Avoid these emphatic claims. I suggest we keep the second sentence (aim) only.

Introduction

Estrogen deficiency in m ADDIN EN.CITE on in hepatocytes, .........There appears something missing here. Authors should 'fix' this sentence

Discussion

Could we have some suggested explanations or discussion on why at AIP> 0.60, we have reduced risk of depression among the postmenopausal cohort. Is it due to chance (I doubt)?, the characteristics of the study cohort? This is pivotal and needs to be discussed.

Authors should revisit the limitation section. Comment on the unavailability of data on estrogen and other endocrine and biomarkers and how they would have impacted findings.....

Reviewers' comments:

Reviewer's Responses to Questions

**Comments to the Author**

1. Is the manuscript technically sound, and do the data support the conclusions?

Reviewer #1: Yes

2. Has the statistical analysis been performed appropriately and rigorously?

Reviewer #1: I Don't Know

3. Have the authors made all data underlying the findings in their manuscript fully available?

Reviewer #1: Yes

4. Is the manuscript presented in an intelligible fashion and written in standard English?

Reviewer #1: Yes

Reviewer #1: Thank you for inviting me to review the manuscript. Author Ming-Guo Xu has submitted a research manuscript on Association between AIP and depression in pre and post menopausal women. The chosen topic gains significance that it is known that globally perimenopausal women are at significant risk of developing Depression. Few comments to consider:

The manuscript is well written and is easy to follow

Introduction:

I would suggest to add relevant reference for increased susceptibility of postmenopausal women to Depression and also Association of AIP and depression among women in general.if any.

There are few typo errors next to reference 3, pl look into that.

Methods:

The location of study may be mentioned in the methods section.

Also elaborate on how the samples are collected.

There is lack of information regarding the potential confounders and how they are handled ( describe the models).

Results:

Overweight and Obesity are known risk of metabolic syndrome and also atherosclerosis. Would like to know the influence of weight/BMI on the results.

Limitations: Though mentioned, would suggest to list and discuss all the limitations of the study including the confounders, information bias, self reporting bias etc.

**Do you want your identity to be public for this peer review?** For information about this choice, including consent withdrawal, please see our Privacy Policy

Reviewer #1: No

---

## [Author Response · Author response to Decision Letter 1]

6 Jun 2025

Dear Editor and Reviewer,

On behalf of all the contributing authors, I would like to express our sincere appreciation of your constructive comments concerning our article entitled Association between Atherogenic Index of Plasma and depression in premenopausal and postmenopausal women: A cross-sectional study (Manuscript ID: PONE-D-25-15440). We have carefully revised the manuscript to enhance its clarity and facilitate the understanding of the readers. Our point-to-point responses are presented in the following. We hope that the revision will satisfactorily address the comments and concerns of the editors and reviewers.

Our manuscript had been revised to meet PLOS ONE's stylistic requirements. According to the requirements of the journal, we remove any funding-related text from the manuscript and include a funding statement in the cover letter. Changes to our manuscript were all highlighted within the document by using yellow padding.

Editor#

General comments

Question: Authors should however improve on the discussion part finding plausible explanations to support the results obtained or otherwise. There are few situations in text that need to be supported by citation/references.

Response: Thanks for the comment. We had added discussion sections and added reference citations. We interpreted the finding that the risk of depression in postmenopausal women was reduced when the AIP was > 0.60.

Abstract

Question: There are some few studies that has explored AIP and depression in postmenopause (see PMID: 39648880 ....... there are few others). Avoid these emphatic claims. I suggest we keep the second sentence (aim) only.

Response: Thanks for the comments and we agree with you. We keep the second sentence only.

Introduction

Question: Estrogen deficiency in m ADDIN EN.CITE on in hepatocytes, .........There appears something missing here. Authors should 'fix' this sentence.

Response: Thanks. We had amended the sentence ‘Estrogen deficiency in menopausal women decreases estrogen receptor α expression in hepatocytes, resulting in altered expression of key enzymes and genes involved in lipid synthesis and catabolism, and increased serum levels of total cholesterol, triglycerides (TG), and low-density lipoprotein (LDL), and decreased levels of high-density lipoprotein (HDL)’.

Discussion

Question 1: Could we have some suggested explanations or discussion on why at AIP> 0.60, we have reduced risk of depression among the postmenopausal cohort. Is it due to chance (I doubt)?, the characteristics of the study cohort? This is pivotal and needs to be discussed.

Response 1: Thanks for the valuable comments. We reviewed the literature on postmenopausal women's susceptibility to depression and the association of AIP and depression, and added relevant content and references where possible. According to the estrogen withdrawal theory, the likely reason for this is that estrogen levels in postmenopausal women decline significantly and fluctuate almost imperceptibly, and their effect on depression is stronger than the effect of atherosclerosis on depression. Another possible reason lies in the limitations of the study sample. In the population of postmenopausal women, the sample size of AIP > 0.6 was only 67, and the large difference in sample size has a non-negligible impact on the results. Due to the specificity of the postmenopausal period, the relationship between AIP and depression in the postmenopausal female population seems to be more complex compared to premenopausal women. In future studies, we should incorporate both estrogens as well as indicators of atherosclerosis to further explore the mechanisms of depression in postmenopausal women.

Question 2: Authors should revisit the limitation section. Comment on the unavailability of data on estrogen and other endocrine and biomarkers and how they would have impacted findings.....

Response 2: Thanks. Because complete years of estrogen data are not available in the NHANES database, we were unable to verify the relationship between estrogen levels and AIP at different menopausal states and whether estrogen levels affect the relationship between AIP and depression.

Reviewer #

Introduction

Question: I would suggest to add relevant reference for increased susceptibility of postmenopausal women to Depression and also Association of AIP and depression among women in general.if any.

Response: Thanks. We reviewed the literature on postmenopausal women's susceptibility to depression and added relevant content and references as possible.

Methods

Question: The location of study may be mentioned in the methods section.

Also elaborate on how the samples are collected.

There is lack of information regarding the potential confounders and how they are handled ( describe the models).

Response: Thanks. We had added the location of study as well as detailed instructions on how to collect samples. We handled how to deal with potential confounders.

Results

Question: Overweight and Obesity are known risk of metabolic syndrome and also atherosclerosis. Would like to know the influence of weight/BMI on the results.

Response: Thanks for the valuable comments. We divided into two groups according to BMI (BMI < 24 kg/m2 and BMI ≥24 kg/m2) and conducted subgroup analyses. We did not find an interaction in premenopausal and postmenopausal women.

Limitations

Question: Though mentioned, would suggest to list and discuss all the limitations of the study including the confounders, information bias, self reporting bias etc.

Response: Thanks for the comments. We list the limitations of the study as much as possible.

Thank you once again for your valuable feedback. I look forward to receiving further comments from you. If you have any further questions or need further discussion, please feel free to contact me at any time.

Yours sincerely,

All authors

---

## [Editor Report · Decision Letter 1]

10 June 2025

Association between Atherogenic Index of Plasma and depression in premenopausal and postmenopausal women: A cross-sectional study

PONE-D-25-15440R1

Dear Dr. %Mingguo_Xu%,

We’re pleased to inform you that your manuscript has been judged scientifically suitable for publication and will be formally accepted for publication once it meets all outstanding technical requirements.

Kind regards,

Emmanuel Kwaku Ofori, PhD

Academic Editor

PLOS ONE
---

## [Editor Report · Acceptance letter]

PONE-D-25-15440R1

PLOS ONE

Dear Dr. Xu,

I'm pleased to inform you that your manuscript has been deemed suitable for publication in PLOS ONE. Congratulations! Your manuscript is now being handed over to our production team.

Kind regards,

on behalf of

Dr Emmanuel Kwaku Ofori

Academic Editor

PLOS ONE